# Computational Methods for Fluid-Structure Interaction Simulation of Heart Valves in Patient-Specific Left Heart Anatomies

Trung Bao Le [1], Mustafa Usta [2], Cyrus Aidun [2], Ajit Yoganathan [3] and Fotis Sotiropoulos [4,*]

1 Department of Civil, Construction, and Environmental Engineering, Biomedical Engineering Program, North Dakota State University, Fargo, ND 58105, USA; trung.le@ndsu.edu
2 Department of Mechanical Engineering, Georgia Institute of Technology, Atlanta, GA 30332, USA; mustafa.usta@me.gatech.edu (M.U.); cyrus.aidun@me.gatech.edu (C.A.)
3 Department of Biomedical Engineering, Georgia Institute of Technology, Atlanta, GA 30332, USA; ajit.yoganathan@bme.gatech.edu
4 Department of Mechanical and Nuclear Engineering, Virginia Commonwealth University, Richmond, VA 23284, USA
* Correspondence: sotiropoulosf@vcu.edu

**Abstract:** Given the complexity of human left heart anatomy and valvular structures, the fluid–structure interaction (FSI) simulation of native and prosthetic valves poses a significant challenge for numerical methods. In this review, recent numerical advancements for both fluid and structural solvers for heart valves in patient-specific left hearts are systematically considered, emphasizing the numerical treatments of blood flow and valve surfaces, which are the most critical aspects for accurate simulations. Numerical methods for hemodynamics are considered under both the continuum and discrete (particle) approaches. The numerical treatments for the structural dynamics of aortic/mitral valves and FSI coupling methods between the solid $\Omega_s$ and fluid domain $\Omega_f$ are also reviewed. Future work toward more advanced patient-specific simulations is also discussed, including the fusion of high-fidelity simulation within vivo measurements and physics-based digital twining based on data analytics and machine learning techniques.

**Keywords:** heart valves; fluid–structure interaction; data fusion

## 1. Problem Formulation

The human heart is subdivided into four chambers (left atria, left ventricle, right atria, and right ventricle) and consists of four valves (mitral, aortic, tricuspid, and pulmonary valves), as shown in Figure 1. The role of these valves is to direct the blood flow through the chambers toward other organs in a unidirectional fashion via a tandem working mode. For a review of the fluid mechanics of blood flows through heart valves, the reader is referred to our previous reviews [1–3].

Due to the need for personalized treatment for valvular diseases, recent advances in numerical modeling have attempted to replicate valvular mechanics to various degrees. With the emergence of precision medicine for heart diseases [4,5], major emphasis has been placed toward developing a complete virtual model of the human heart [6] by coupling the tissue mechanics and hemodynamics in patient-specific anatomies [7]. The success of anatomically and physiologically realistic digital twin models of the human heart will enable the virtual tests of medical operations that cannot be performed clinically, as well as the development of patient-specific heart valve prostheses and clinical treatments. For that, the patient-specific modeling of heart valves has been increasingly at the forefront of current research [3,8]. This review summarizes recent advances in numerical methods for simulating native and prosthetic valves in the human left heart due to their prevalent risks

of failure. Therefore, only two valves are of interest: the mitral and the aortic valve. For a complete review of heart valve anatomy, the reader is referred to a previous review [9].

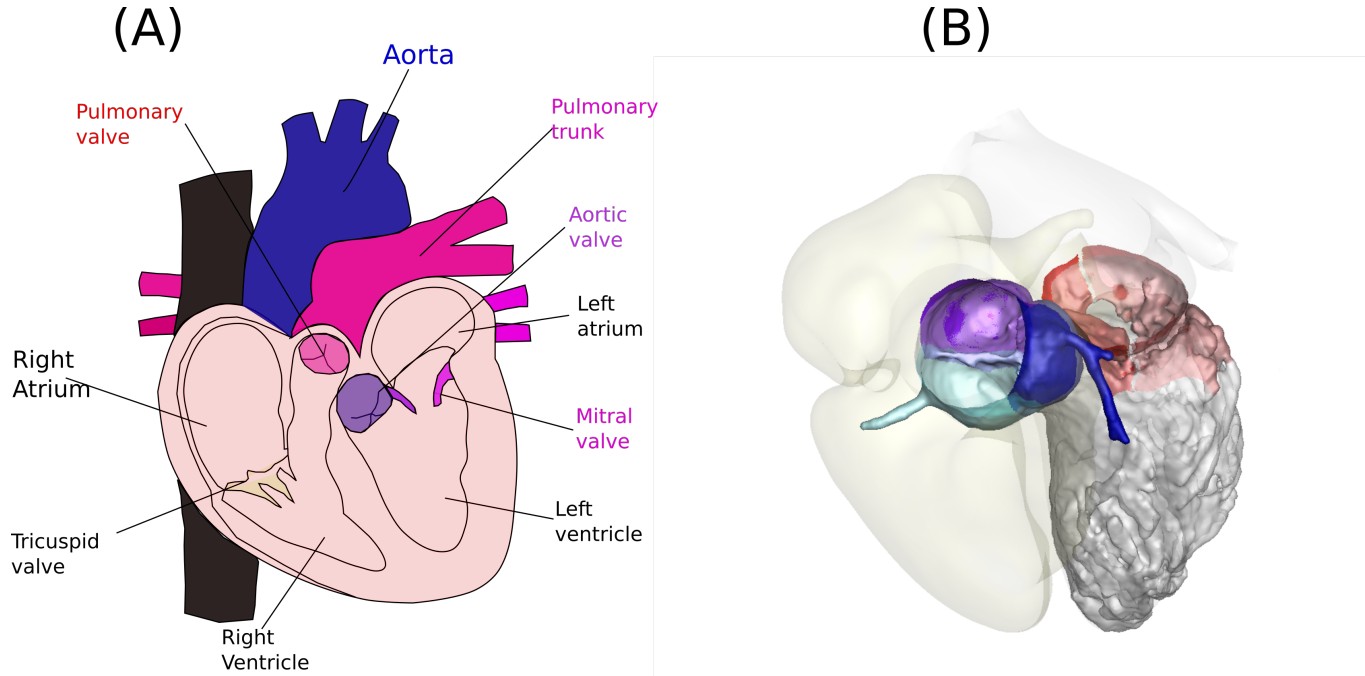

**Figure 1.** (**A**) The anatomy of the left heart including the left atrium, left ventricle, aortic valve, mitral valve, and the sinuses. (**B**) The sinuses and the valve leaflets are shown in colors to emphasize their anatomical location: (i) left coronary artery (LCA); (ii) right coronary artery (RCA); and (iii) non-cusp coronary artery (NCA). The mitral leaflets are shown in red. The right ventricle is shown in the shadow to highlight the orientation of the left heart. The ventricular structures are visualized using the public computed tomography data of a human subject, which can be found at: http://www.gimias.org/index.php?option=com_content&view=article&id=26&Itemid=18 (accessed on 15 August 2021).

Aortic and mitral valves are thin multi-layer tissue structures [10]. This feature ensures a complex response of the valve under mechanical loading. As shown in Figure 1, the aortic valve consists of three thin cusps (LCA, RCA, and NCA). The mitral valve consists of two leaflets (anterior and posterior leaflets) and the papillary muscles. Over the years, computational methods for the structural dynamics of native aortic and mitral valves have been developed to various levels of sophistication. The aortic valve [11] and mitral valve [12] dynamics are highly complex, as the leaflets interact with blood flows under the influence of anatomical features, such as the sinuses [13,14] or the myocardial surfaces [15]. A high-fidelity simulation of heart valves [2] is thus required to resolve small-scale dynamics, such as fluttering [11].

The interaction of the blood flow with heart valves leads to highly intricate flow structures within the heart chambers, such as shear layer roll-up, vortex ring formation, breakdown, and transition to turbulence [3]. Vortex ring formation induced by shear-layer roll-up [16–18] has been shown to increase viscous dissipation [19,20] and could also lead to high shear stresses [21], increasing the risk for thrombus formation. Regions of flow stagnation inside the sinuses could also lead to thrombosis [22] or leaflet calcification [23] due to the associated low-shear-stress environment. Numerical methods for heart valves [24] must be able to account for the large deformation of valvular leaflets as well as the aforementioned complex flow characteristics. Numerical methods capable of handling such complexities are the subject of this review.

Conceptually, the left heart system consists of tissues and blood flow. The tissues include the epicardium, myocardium, endocardium, mitral/aortic valves, left atrium, and

aorta. With reference to Figure 2, we shall denote in the remainder of this review such tissues as $(\Omega_s)$, and the blood as the fluid domain $(\Omega_f)$. Blood is a complex fluid with about 45% highly deformable red blood cells suspended in a Newtonian fluid (plasma). The rheological characteristics, including the normal stress and shear viscosity, have been investigated in detail [25]. As it is currently prohibitively expensive to resolve individual red blood cells for valvular flow, blood is considered an incompressible Newtonian fluid in this case. This assumption is accurate for simulations in the left heart where the size of the red blood cells is many orders of magnitude smaller than the dimensions of the flow domain [26]. In regions where these assumptions break down, other methods that can account for the cellular nature of blood need to be employed [27].

The interface between the fluid and the solid domain is denoted as $\Gamma = \partial\Omega_f = \partial\Omega_s$. The portions of the interface between the valvular leaflet interface and the blood flow is denoted as $\Gamma_{FSI}$, since the motion of the leaflets is typically determined via a coupled fluid–structure interaction (FSI) algorithm. The endocardium surface, the atrial inlets, the aorta, and the outlet of the descending aorta are denoted as $\Gamma_{LV}$, $\Gamma_{inlet}$, $\Gamma_{aorta}$, and $\Gamma_{outlet}$, respectively. Therefore, in the computational domain the interface $\Gamma$ between solid and fluid is given by $\Gamma = \Gamma_{FSI} \bigcup \Gamma_{LV} \bigcup \Gamma_{inlet} \bigcup \Gamma_{aorta} \bigcup \Gamma_{outlet}$, as shown in Figure 2.

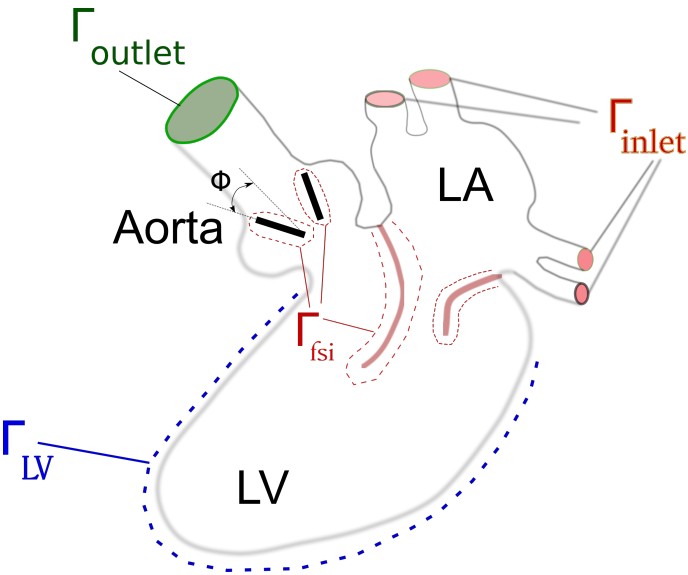

**Figure 2.** The configuration of the computational domain. The fluid domain $(\Omega_f)$ of the left heart is divided into three main regions: (i) the left atrium (LA); (ii) the left ventricle (LV); and (iii) the aorta. The motion of the left ventricle is tracked on the boundary $\Gamma_{LV}$. The valve motions $(\Omega_s)$ are computed from the fluid–structure interaction algorithms by exchanging the kinematics and loading conditions on $(\Gamma_{fsi})$. Blood flow comes from the lung via the inlets at $\Gamma_{inlet}$ and exits at the outlet $\Gamma_{outlet}$. The motion of the mechanical valve is tracked with the opening angle $\phi$.

The aortic domain $(\Gamma_{aorta})$ may be simplified to be stationary, or its motion may be prescribed from patient-specific imaging data, or in most general models, may also be part of the FSI domain. The flow rate at the $\Gamma_{outlet}$ is assumed [28] to follow several known laws (e.g., Windkessel model) that describe the hydraulic resistance of the rest of the cardiovascular system [29]. The motion of the left ventricle $\Gamma_{LV}$ is known via non-invasive modalities [30] or is computed from the tissue modeling [31]. The flow inlets at the left atrium $\Gamma_{inlet}$ are prescribed using patient-specific data [18,30,32]. In brief, the left heart domain consists of parts that move either with prescribed motion or due to coupled non-linear FSIs. The fluid–solid interface $\Gamma$ can thus be expressed as follows: $\Gamma = \Gamma_M \bigcup \Gamma_s$, where $\Gamma_M$ is the moving portion of the boundary $(= \Gamma_{FSI} \bigcup \Gamma_{LV})$ and $\Gamma_N$ is the portion of the boundary that is held stationary $(= \Gamma_{inlet} \bigcup \Gamma_{aorta} \bigcup \Gamma_{outlet})$.

The interface between the solid and the fluid domain is discretized using a set of material points [33] $i = 1, I$ with coordinates $\mathbf{x_i}$ defining the interface $\Gamma = \Gamma(\mathbf{x_i})$. The

motion of material points that are part of $\Gamma_M$ are tracked in a Lagrangian manner by solving the following equation:

$$\mathbf{v}_i = \frac{d\mathbf{x}_i}{dt} \quad \forall \mathbf{x}_i \in \Gamma_M \tag{1}$$

where $\mathbf{v}_i$ is the velocity vector of the $i$th material point. The two portions, $\Gamma_{FSI}$ and $\Gamma_{LV}$, of $\Gamma_M$ move as a result of different physical processes, and each one of them is treated with different numerical techniques. The details of $\Gamma_{FSI}$ kinematics are discussed in Section 4. Our main effort in this review is to focus on the algorithms to obtain the kinematics of $\Gamma_{FSI}$, as described below.

## 2. Prosthetic Heart Valves

A large number of patients need heart valve replacement every year worldwide due to the failure of their native heart valves [34]. Recent advances in manufacturing have led to an explosive growth in the types of heart valve prosthesis [35] as biomaterial design has advanced rapidly [36]. In this work, these prostheses are categorized into: (i) mechanical valves and (ii) bioprosthetic valves, which emulate the shape and biological deformation patterns of native valves. For a detailed hemodynamic analysis of each type, the reader is referred to previous reviews [1,3].

### 2.1. Mechanical Valves

The implantation of a heart valve prosthesis at the left ventricle outflow tract (LVOT) is one of the most common procedures when the native aortic tricuspid valve malfunctions (see Figure 2). However, mechanical prostheses for the mitral valve [37] are also implanted in many patients. Mechanical valves remain a viable choice among other options due to their long durability [38].

The mechanical valves are considered as rigid prostheses with moving parts [39]. Three types of mechanical valves have been proposed: (1) single leaflet; (2) bi-leaflet (BMHV); and (3) tri-leaflet valves. While the bi-leaflet valve is the most common type [21], the emergence of the tri-leaflet mechanical heart valve [40,41] is also important since it is aimed at reducing thrombogenicity. In mechanical valves, the leaflets are rigid and non-deformable. For the purpose of demonstration, the BMHV is represented in Figure 2 at the left ventricle outflow tract (LVOT). Two rigid leaflets pivot around the hinge under a blood pressure wave from the contraction of the left ventricle during systole. It is now well understood that the simple design of BMHV induces non-physiologic flow patterns to form at the aortic root due to the opening and closing of the two leaflets [1,21]. Therefore, understanding the hemodynamic environment induced by the prosthetic valve is critical to improve the valve design and implantation procedure [42–45].

The motion of the two leaflets is a rigid body rotation around their axes of rotation. As shown in Figure 2, $\phi$ is denoted as the opening angle of the leaflet, which can be used to express the position vector ($\mathbf{x}(X, Y, Z)$) of a material point on the leaflet as follows:

$$\mathbf{x} - \mathbf{x}_c = R(\phi)\mathbf{r_c} \tag{2}$$

where $\mathbf{x_c}(X, Y_c, Z_c)$ is the projection of the material point on the rotational axis, $R(\phi)$ is the in-plane rotational matrix (see below). To simplify the discussion, we assume that the rotational axis aligns to the $X$ direction, and the radial distance to the rotational axis ($\mathbf{r_c}$) can be computed as: $|r_c| = \sqrt{(Y - Y_c)^2 + (Z - Z_c)^2}$.

Typically, the leaflets are constrained to open and close up to a certain value of the opening angle. For example, the maximum angle $\phi_{max} = 58^0$ (fully closed) and the minimum angle is $\phi_{min} = 5^0$ (fully open). The in-plane (i.e., $X = const$) rotational matrix $R(\phi)$ is defined as follows:

$$R(\phi) = \begin{bmatrix} 1 & 0 & 0 \\ 0 & \cos(\phi) & -\sin(\phi) \\ 0 & \sin(\phi) & \cos(\phi) \end{bmatrix} \tag{3}$$

The governing equation of the leaflet motion is obtained from the conservation of angular momentum and can be written in terms of $\phi$ as follows:

$$\mathbf{I_0}\frac{d^2\phi}{dt^2} = \mathbf{M_0} \tag{4}$$

Here $\mathbf{I_0}$ is the reduced moment of inertia, which is calculated as:

$$\mathbf{I_0} = \frac{\rho_s \int_{\Omega_s} |\mathbf{r_c}|^2 dV}{\rho_f D^5} \tag{5}$$

where $\rho_s$ and $\rho_f$ are the specific weight of the solid and fluid, respectively. Finally, $\mathbf{M_0}$ is the moment coefficient:

$$\mathbf{M_0} = \frac{\mathbf{M_X}}{\rho_f U^2 D^3} \tag{6}$$

Because the leaflets are assumed to rotate around the rotational axes (which are parallel to the $X$ axis), we can compute the moment around the rotational axis found by integrating the fluid traction vector $\mathbf{t_f}$ on the interface $\Gamma_{FSI}$:

$$\mathbf{M_X} = \int_{\Gamma_{FSI}} \mathbf{r_c} \times \mathbf{t_f} d\mathbf{A} \tag{7}$$

Assuming that the angular position $\phi$ and angular velocity $\omega = d\phi/dt$ of the leaflet are known at timestep $n$, it is necessary to find the position at $n+1$ via Equation (4) via an implicit procedure. To solve Equation (4), pseudo time stepping using $\triangle\tau$ [26] is typically used to find the angular velocity $\dot{\phi}^{n+1}$ with looping variable $l$:

$$\frac{\dot{\phi}^{l+1} - \dot{\phi}^l}{\triangle\tau} + \frac{3\dot{\phi}^l - 4\dot{\phi}^n + \dot{\phi}^{n-1}}{2\triangle t} = M^{n+1} \tag{8}$$

After the angular velocity $\dot{\phi}^{n+1}$ is found, the angle $\phi$ can be updated by midpoint rule [46] as:

$$\phi^{n+1} = \phi^n + \triangle t \frac{\dot{\phi}^n + \dot{\phi}^{n+1}}{2} \tag{9}$$

The structural solver therefore can be written as an operator estimating the position vector $\mathbf{x}$ (and thus the angle $\phi$) from the external load $\mathbf{M}$ and boundary conditions on $\Gamma_{FSI}$ as follows:

$$\phi = \mathbb{S}(\Gamma_{FSI}, \mathbf{M}) \tag{10}$$

### 2.2. Bioprosthetic Valves

Bioprosthetic valves are the most advanced valvular prostheses [35] to date. The structural components of those prostheses consist of thin leaflets and flexible stents, while the structural mechanics [47] of these valves are well studied, their long-term interaction with blood flows [22] remains a concern due to the risk of thrombogenicity [48]. In this review, we focus on the interaction of leaflets and blood flow dynamics. The impact of stents and deployment strategies [31,49,50] are not considered in this review.

Since the leaflets largely deform, it is essential to track all points on the leaflet domain $\Omega_s$. Here we use the Lagrangian framework to describe the motion of material points. The current position $\mathbf{x}$ of a material point at time $t$ is related to its position at the reference configuration $\mathbf{X}$ by the mapping $\mathbf{\Psi}$:

$$\mathbf{x} = \mathbf{\Psi}(\mathbf{X}) \tag{11}$$

The gradient tensor of the transformation is therefore:

$$\underline{\underline{F}} = \frac{\partial \mathbf{\Psi}}{\partial \mathbf{X}} \tag{12}$$

The displacement of a material point is defined as:

$$\mathbf{u} = \mathbf{x} - \mathbf{X}, \tag{13}$$

and the velocity of the material point is given as:

$$\mathbf{v_s} = \dot{\mathbf{u}} = \mathbf{du}/\mathbf{dt} \tag{14}$$

The momentum equations for the solid part, formulated in the current configuration, have the following form:

$$\rho_s \frac{d\dot{\mathbf{u}}}{dt} = \nabla \cdot \underline{\underline{\sigma_s}} + \rho_s \mathbf{b}, \tag{15}$$

where $\rho_s$ is the current mass density of the material. Here, $\sigma_s$ is the Cauchy stress tensor, which represents the gradient operator in the current configuration, and $\mathbf{b}$ is the body force per unit mass.

The boundary of the solid structure can be further represented as the sum of non-overlapping parts $\partial \Omega_s = \Gamma_s^N \bigcup \Gamma_s^D \bigcup \Gamma_{fsi}$, where the indices $D$ and $N$ denote boundaries with Dirichlet and Neumann conditions, respectively:

$$\begin{aligned} \dot{\mathbf{u}} &= \bar{\mathbf{u}} && \text{on} \quad \Gamma_s^D, \\ \underline{\underline{\sigma_s}} \cdot \mathbf{n}_s &= \bar{\mathbf{t}}_s && \text{on} \quad \Gamma_s^N, \end{aligned} \tag{16}$$

where $\Gamma_s^D$ and $\Gamma_s^N$ represent the portions of the surface of the body in its current configuration where Dirichlet and Neumann conditions are applied, respectively. $\bar{\mathbf{u}}$ and $\bar{\mathbf{t}}_s$ are known velocity and traction vectors acting on the surface. $\mathbf{n}_s$ is a unit normal to the boundary and $\bar{\mathbf{u}}$ is the velocity prescribed on the surface.

In the fluid–structure interaction, additional boundary conditions must be implemented on the $\Gamma_{fsi}$:

$$\begin{aligned} \dot{\mathbf{u}} &= \mathbf{v_f} && \text{on} \quad \Gamma_{fsi}, \\ \underline{\underline{\sigma_s}} \cdot \mathbf{n}_s &= \mathbf{t}_f, && \text{on} \quad \Gamma_{fsi}, \end{aligned} \tag{17}$$

here, $\Gamma^{fsi}$ is part of the moving structure surface, the configuration of which needs to be determined by solving the FSI problem, $\mathbf{t}_f = \underline{\underline{\sigma_f}} \cdot \mathbf{n}_f$ is a traction vector which acts on this part of the surface from the fluid, $\underline{\underline{\sigma_f}}$ and $\mathbf{n}_f$ are the stress tensor and surface normal unit vector from the fluid domain $\Omega_f$.

The solid momentum equations with the boundary conditions can be reformulated in terms of an operator $\mathbb{S}$, which incorporates both the (kinematic and dynamic) boundary conditions and constitutive equations to yield the velocity $\dot{\mathbf{u}}$ and displacement field $\mathbf{u}$:

$$\mathbf{u} = \mathbb{S}(\Gamma_{fsi}, \mathbf{t}_f, \mathbf{v_f}) \text{ in } \Omega_{\mathbf{s}}, \tag{18}$$

here $\mathbf{v_f}$ and $\mathbf{t}_f$ are applied at the boundary $\Gamma_{fsi}$.

## 3. Fluid–Structure Interaction of Valves and Blood Flows

The kinematics of the leaflets are the result of the interaction between the blood flow dynamics and the inertia of the leaflets. The fluid–structure interaction (FSI) procedure thus requires the coupling of the fluid solver $\mathfrak{F}$ (for the domain $\Omega_f$) and the structural solver $\mathbb{S}$ (for the domain $\Omega_s$). The goal of the FSI algorithm is to find the value of the angle $\phi$ (Equation (10)) or the displacement $\mathbf{u}$ (Equation (18)). The loading condition $\mathfrak{L}$ (either the moment $\mathbf{M}$ or the traction $\mathbf{t_f}$) is calculated from the fluid solver ($\mathfrak{F}$). This loading $\mathfrak{L}$ is

prescribed as a Neumann boundary condition for the structure solver ($\mathbb{S}$). The structural solver is then used to find the position of the leaflets, which is prescribed as Dirichlet boundary conditions for the fluid solver ($\mathfrak{F}$). Note that the kinematic condition requires the continuity of the interface between the solid and fluid domains:

$$\Gamma_s \equiv \Gamma_f \equiv \Gamma_{FSI} \tag{19}$$

Note that the solid/fluid boundary, which consists of the fluid–structure interaction interface $\Gamma_{FSI}$, is also a function of the vector position $\mathbf{x}$. Therefore:

$$\Gamma_{FSI} = \Gamma(\mathbf{x}) \tag{20}$$

The dynamic condition requires the continuity of the stress at the interface:

$$\sigma_f = \sigma_s \tag{21}$$

with a no-slip condition on the interface $\Gamma_{FSI}$:

$$\mathbf{v}_s = \mathbf{v}_f \tag{22}$$

The governing equations for the fluid ($\mathfrak{F}$) and solid ($\mathbb{S}$) can now be combined and expressed in an operator form as follows:

$$\mathfrak{L} = \mathfrak{F}(\Gamma(\mathbf{x})) \tag{23}$$
$$\mathbf{x} = \mathbb{S}(\Gamma(\mathbf{x}), \mathfrak{L}) \tag{24}$$

Note that the operators $\mathbb{S}$ and $\mathfrak{F}$ change with time and are dependent on the initial and boundary conditions imposed on the boundary $\Gamma$. Thus, this system of equations can be written in compact notation:

$$\mathbf{x} = \mathbb{S} \circ \mathfrak{F}(\mathbf{x}) \tag{25}$$

where the operator $\circ$ denotes the transfer load at the interface $\Gamma_{FSI}$ from the fluid solver to the solid solver and supply for the solid solver $\mathfrak{S} = \mathbb{S}(\mathfrak{L})$. Therefore, the coupling between the solid solver $\mathbb{S}$ and the fluid solver $\mathfrak{F}$ is equivalent to finding the fixed point of the operator $\mathbb{S} \circ \mathfrak{F}$.

Solving Equation (25), however, is challenging. The pulsatile nature of the flow waveform [51] imposes a sharp increase in the pressure gradient across the valve. In addition, the low-density ratio between the leaflet material and blood poses another challenge to find the root of Equation (25) [26]. Many works have addressed this issue [52,53]. One popular approach is to use a fixed-point iteration (see the Algorithm 1) with relaxation [26,54].

To demonstrate the algorithm, assuming that the leaflet angle $\phi$ in Equation (10) is known at time step $n - 1$, Equation (25) is solved to obtain the leaflet angle at timestep $n$ with the current boundary conditions on $\Gamma$ via a series of strong-coupling sub-iterations [26]. The Aitken non-linear relaxation technique is used to accelerate convergence and enhance robustness [26,55]. The details of the algorithm for each time step $n$ are as follows:

---

**Algorithm 1:** Fixed-point iteration algorithm to solve the Equation (25)

---

$l = 0$
$\phi^0 = \phi_{n-1}$
**while** $|\phi^l - \phi^{l-1}| > tolerance$ **do**
$\quad l = l + 1$
$\quad \widetilde{\phi^{l+1}} = \mathbb{S} \circ \mathfrak{F}(\phi^l)$
$\quad e^l = \widetilde{\phi^{l+1}} - \phi^l$
$\quad \phi^{l+1} = \lambda^l \widetilde{\phi^{l+1}} + (1 - \lambda^l)\phi^l$
**end while**
$\phi_n = \phi^{l+1}$

---

Here, the new guess $\phi^{l+1}$ is found by one under-relaxed Richardson iteration of $\widetilde{\phi^{l+1}}$, with the relaxation factor calculated by the Aitken accelerator [26,55]:

$$\Delta e^l = e^l - e^{l-1} \tag{26}$$

$$\lambda^l = -\lambda^{l-1}\frac{e^{l-1}}{\Delta e^l} \tag{27}$$

Note that in Equation (27), the recursive nature of $\lambda^l$ enables the current guess $\phi^l$ to implicitly link to all previous sub-iterations. Since the first sub-iteration $l = 1$ the previous residual $e_0$ is not available, and a pre-determined value of $\lambda^1$ must be used. A popular choice of $\lambda^1$ is 0.7. The stability criteria for $\lambda^l$ are discussed in reference [26]. The implementation of this fixed-point iteration in the context of the deformable valve is done in a similar fashion [54] for the position vector **x** [56]. However, the convergence property of the fixed-point equation depends strongly on the structural characteristics of the leaflets [52]. The reader is referred to our previous review [53] on this coupling matter.

One critical issue of this coupling occurs during valve closure [57]. As blood flow reverses its direction, the pressure gradient increases exponentially [58]. This adverse pressure gradient drives the valve leaflets to coapt [59]. Resolving the dynamics of each leaflet requires solving Equation (25) accurately [60]. The retrograde flow from the valve closure can reach far from the valve location [61] and destabilizes the left ventricular flow field [60]. The retrograde flow might play a critical role in modifying flow patterns during the filling phase (E-wave) [62].

## 4. Patient-Specific Anatomy and the Dynamics of $\Gamma_{LV}$

Recent advancements of non-invasive measurement techniques and numerical methods gave rise to the emerging field of patient-specific modeling (PSM) [63]. This type of modeling is the combination of state-of-the-art numerical simulations and the in vivo measurement data. PSM utilizes all individualized geometry information, such as anatomical and ultrasound data from non-invasive imaging techniques (magnetic resonance imaging (MRI) [18] or computed tomography (CT) [30]), to calculate the hemodynamic environment within the left heart [18,30,32]. Hemodynamics inside the left heart can now be simulated using patient-specific data, providing unique opportunities for disease diagnosis or the treatment of individuals [31]. Virtual surgery with different surgical scenarios [64] could be tested before the actual operation, thus, helping surgeons evaluate a wide range of options prior to entering the operation room [50]. Therefore, the development of versatile and efficient numerical methods for solving the patient-specific hemodynamic problems, especially problems involving fluid–structure interaction with prosthetic valves, remains a frontier research problem and is at the center of much of the ongoing research in the field today [7,54,65].

Understanding the patient-specific hemodynamics of heart valves as illustrated in Figure 3, however, requires good quality anatomical and hemodynamics data [66]. As shown in Figure 2, the motion of the left ventricle $\Gamma_{LV}$ drives the pressure gradient within the LV chamber and across the aortic/mitral valves. The present-day scanning frequency per cardiac cycle (frames/s) of various imaging modalities is technologically limited. For example, the four-dimensional-flow magnetic resonance (4D-flow MRI) imaging technique [67] provides useful three-dimensional-flow fields inside the ventricular chambers during the entire cardiac cycle, as shown in Figure 3. Nevertheless, 4D-flow MRI is typically constrained by relatively coarse spatial ($\approx$3 mm $\times$ 3 mm $\times$ 3 mm) and temporal (30–50 ms) [68] resolutions. The availability of computed tomography images for patients [30] is sparse since CT is not approved to be used in all patients. Ultrasound data is available [69–71] with a higher temporal resolution, but it is challenging to reconstruct the left ventricular 3D kinematics from individual images [72]. Therefore, it is essential to perform data interpolation between successive scanned images to reconstruct the left ventricular wall motion ($\Gamma_{LV}(t)$) over the cardiac cycle with the resolution of choice [18]. The accuracy

of the resulting kinematics, and consequently the clinical relevance of the valvular simulation, depends both on the accuracy of the interpolation technique and the initial temporal resolution of the scanned images.

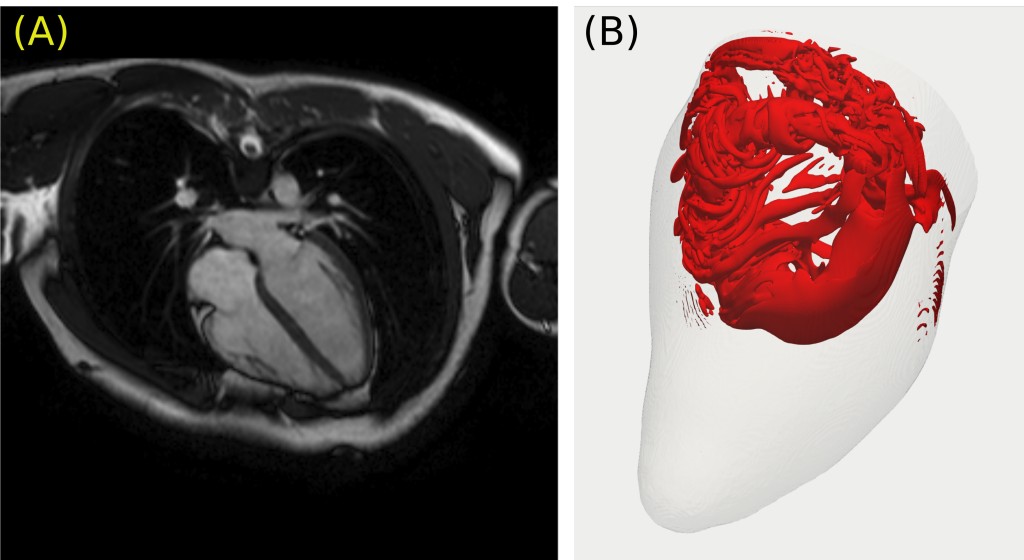

**Figure 3.** The complexity of left ventricular flow [18]. The motion of the heart is monitored using magnetic resonance imaging in (**A**). The numerical simulation using the prescribed kinematics ($\Gamma_{LV}(t)$) is shown in (**B**). The mitral valve is not included in the simulation. Instead, the 4D-Flow MRI data is used to prescribe the incoming jet from the left atrium. The mitral vortex ring (MVR) forms during early diastole (E-wave) as the jet accelerates. The MVR is visualized using Q-criterion [73].

Following the reconstruction procedure of $\Gamma_{LV}(t)$, many works have been able to perform patient-specific simulations of left ventricular flows [18,32,74]. The high-resolution simulation of an implanted heart valve prosthesis in a patient-specific beating left heart [60] has demonstrated the importance of resolving the left ventricular flow accurately [75]. For a review of patient-specific ventricular simulation, the reader is referred to other recent works [65].

## 5. Continuum Approaches for Solving the Flow Dynamics in $\Omega_f$

### 5.1. Governing Equations for the Fluid Domain $\Omega_f$

The continuum approach is the most popular approach for simulating intraventricular flows [65]. Here, blood is treated as an incompressible Newtonian fluid with constant viscosity $v = 3.33 \times 10^{-6}$ m$^2$/s and a specific weight $\rho_f = 1050$ kg/m$^3$.

The blood motion is governed by the unsteady, three-dimensional Navier–Stokes equations:

$$
\begin{aligned}
\nabla \cdot \mathbf{v_f} &= 0 \\
\frac{\partial \mathbf{v_f}}{\partial t} + \nabla \cdot (\mathbf{v_f} \otimes \mathbf{v_f}) &= \nabla \cdot \tau
\end{aligned}
\tag{28}
$$

where the stress tensor $\tau$ relates to the pressure $p$ and strain rate $\epsilon$: $\tau = -p\mathbf{I} + 2\mu\epsilon(v_f)$ and $\epsilon(v_f) = \frac{1}{2}(\nabla v_f + (\nabla v_f)^T)$, $\mu = \rho_f v$. The notation $\otimes$ denotes the tensor product of two vectors.

To solve Equation (28), boundary conditions must be specified on the solid/fluid interface $\Gamma$. As seen in Figure 2, $\Gamma$ consists of solid surfaces that are either stationary or moving, as well as inflow and/or outflow boundaries resulting from truncating the connection of the LV/aorta system from the rest of the cardiovascular system. Depending on the characteristics of the boundary portion, different strategies are implemented to reconstruct the boundary conditions.

At the mitral inlets $\Gamma_{inlet}(t)$ (see Figure 2), the time-dependent blood flow fluxes from the lung are typically prescribed from measurements [32,76]. The mitral valve dynamics can be prescribed from measurements [32,76,77] or computed from FSI algorithms [78,79]. Note that it is challenging to obtain the patient-specific anatomy of the mitral valve as well as its mechanical properties due to the limitation in imaging technologies [30].

Outflow boundary conditions need to be imposed at the outflow of the aortic flow track $\Gamma_{outlet}$. The flux into the descending aorta $Q_{outlet}$ results from the difference between the mitral flux $Q_{inlet}$ and the volume flow rate of change of the LV chamber. That is:

$$Q_{outlet}(t) = Q_{inlet}(t) - \frac{d\mathfrak{V}_{LV}}{dt} \tag{29}$$

This condition must be satisfied at all times for a well-posed problem. Special techniques have been developed to ensure this condition at the outlet [28,29,80].

Along the $\Gamma_{LV}$ portion of the boundary, the time-dependent LV wall motion ($\Gamma_{LV}(t)$), obtained with the reconstruction methodology in Section 4, is prescribed as the input to the simulation and used to drive the LV blood flow [17,77,81]. The no-slip and no-flux boundary conditions are imposed for the velocity field at the LV wall portion $\Gamma_{LV}$ as follows:

$$\mathbf{v_f}(t) = \mathbf{v_s}(t) \text{ on } \Gamma_{LV} \tag{30}$$

The motion of the aortic/mitral valve leaflets is driven by the beating left ventricle and, thus, the velocity at the interface between the valve leaflets and the blood flow ($\Gamma_{FSI}$) needs to be obtained by a coupled FSI procedure, as discussed in Section 3. In the following sections, the algorithms solving the Navier–Stokes equations (Equation (28)) in $\Omega_f$ will be discussed. Since the motion of $\Gamma_{LV}$ and $\Gamma_{FSI}$ poses a great challenge for numerical methods, different approaches have been proposed to deal with the deformation of $\Omega_f$ and $\Omega_s$, as well as the overlapping region $\partial\Omega_f \bigcap \partial\Omega_s$.

### 5.2. Finite Difference Methods

Fixed-mesh methodsfor cardiovascular flow first emerged in the 1970s when the immersed boundary method (IBM) was introduced by Peskin for heart simulation problems [82,83]. In this method [7], a fixed background mesh for the fluid solver is used in the entire computational domain while the motion of solid immersed boundaries is presented by including a force field in the right-hand side of the Navier–Stokes equations. The solid body is therefore implicitly removed from the computational domain. The fluid solver only "sees" its existence through the layer of a near-solid surface called "immersed nodes" (IB nodes). The added force is distributed via a discrete delta-function over several grid nodes surrounding the solid surface, and as a result, the solid/fluid interface is smeared across these grid points. Because of this inherent smearing feature, the original IB method is known as a diffused interface method. It is only first-order accurate in space and requires an adaptive mesh refinement to achieve higher accuracy [59].

In order to solve the smearing of the interface problem, a new class of IB methods called "sharp interface IB methods" (SIB) [84] have been introduced. The main distinction between the SIB method and the original IB method is the representation of the interface. In SIB methods, the interface is reconstructed, and its velocity is directly specified or "forced". Thus, the most important part of SIB methods is the method used to reconstruct the velocity field at the IB nodes. Recent works focus on the adaptive mesh refinement techniques to enhance the local resolution near the wall [85]. Due to the incompressibility constraint ($\nabla \cdot \mathbf{v}_f = 0$), the staggered grid approach [54,59,86] is typically implemented to ensure that the divergence-free condition is satisfied exactly at the cell center (pressure node). The mass flux of the face center is thus stored separately. For details of the sharp-interface method for cardiac flows, the reader is referred to a recent review [65].

The sharp-interface immersed boundary method has been applied successfully for many aortic valve applications [42,43,54,87–90] as well as the intraventricular flows with

both valves [79]. To demonstrate the capability of the method, a simulation of a tri-leaflet valve [91] is shown in Figure 4. The evolution of the flow structure is rather complicated [92], with the formation of the leading vortex ring and its breakdown [3,89]. This process is explained further in Figure 4, where the leading ring in Figure 4A is created by the acceleration of the peak systolic flow, which opens the valve's leaflet. The shear layer instability is shown in Figure 4B and leads to the transition to turbulence in Figure 4C. Here, the numerical simulation is able to resolve up to the spatial resolution of ≈100 µm. Such a resolution is required [3] to evaluate blood cell damage and platelet activation [42]. This is an important issue in pathological conditions [93] that can induce complex flows in the aortic sinus and the ascending aorta, as shown in Figures 5 and 6.

The next frontier in simulating cardiac flows is the ability to capture the dynamics of both valves accurately [7] with patient-specific anatomy. Despite the importance of the mitral valve to the left ventricular hemodynamics [94], the application of immersed boundary methods for the mitral valve has a lesser extent of success [95]. The main challenge is the complex anatomy of the mitral valve itself [96]. In addition, obtaining the necessary structural information for the mitral valve is a challenging task, even with the most sophisticated technology to date [97].

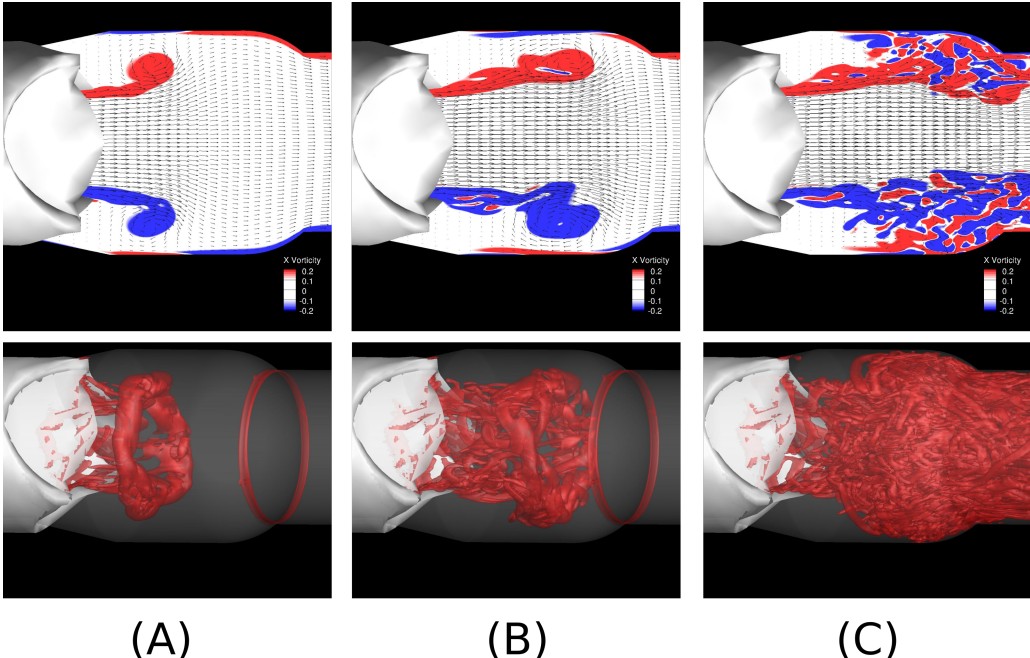

**Figure 4.** Fluid–structure interaction of a tri-leaflet valve in an axisymmetric aorta [91] with the peak systolic Reynolds number of 2580, which is based on the peak systolic velocity $U = 0.78$ m/s and the diameter of the valve $D = 25.4$ mm. The figures respectively show the time instantsat (**A**) $t_A = 200$ milliseconds, (**B**) $t_B = 260$ milliseconds and (**C**) $t_C = 380$ milliseconds, respectively. The top row shows the evolution of the leading vortex ring, which is visualized by the contour of out-of-plane vorticity. The bottom row shows the three-dimensional structures, which are visualized by the iso-surface of Q-criterion [73].

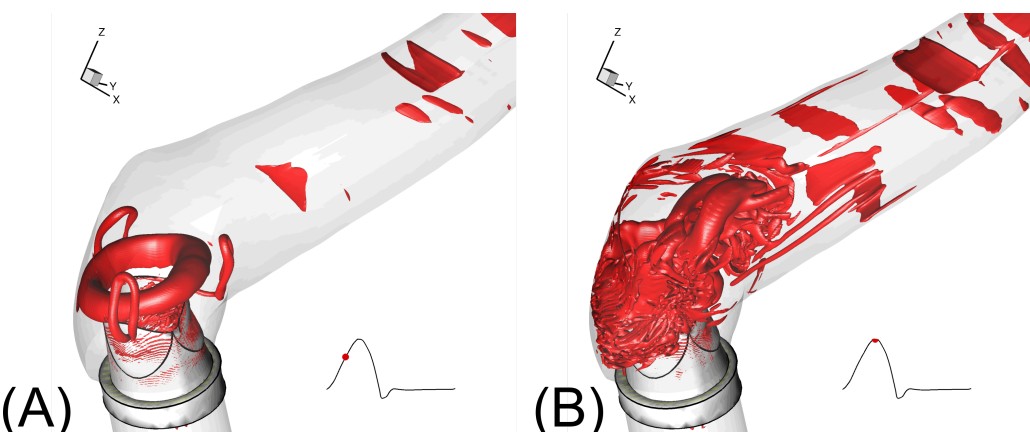

**Figure 5.** Fluid–structure interaction of a tri-leaflet valve in a patient-specific aorta [98]: (**A**) early systole and (**B**) peak systole. The evolution of the leading vortex ring is visualized by the iso-surface of Q-criterion [73].

Moving-mesh methods employ a dynamic deforming mesh that conforms with and remains attached to the solid surface at all times. In cardiovascular flows, arbitrary Lagrangian–Eulerian formulation (ALE) has been widely used for a long time [99]. In this method, the interface between solid and fluid is tracked. Because the computational mesh continuously deforms to conform with the moving interface, large structural deformations can cause severe distortions of the mesh. In such cases, frequent re-meshing is required [100]. The ALE capability has been developed for commercial software. These are widely popular for valvular applications [101–104]; however, while the ALE method has been demonstrated successfully for the aortic valve [105,106] and the left ventricle [100], there have been no attempts to simulate both the aortic and mitral valves simultaneously using ALE.

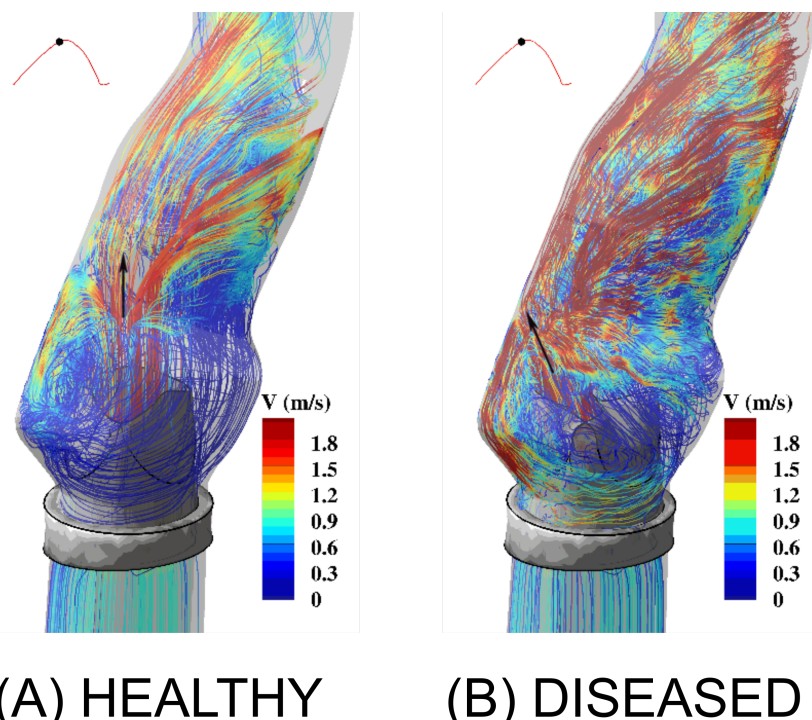

## (A) HEALTHY　　(B) DISEASED

**Figure 6.** Flow patterns at the aortic sinus under healthy (tricuspid) and diseased (bicuspid) aortic valves. The flow is visualized by streamlines colorized by velocity magnitude. The black arrow indicates the impingement location of the aortic jet. Reprinted by permission from *Springer Theoretical and Computational Fluid Dynamics* [88].

### 5.3. Immersed Finite Element Methods

The challenge in the moving-mesh method has motivated the development of embedded methods [107] in the last few years. This class of methods essentially replicates the capability of the immersed boundary method using finite element approximation [108]. In this method, the presence of the thin valvular structures is described independently from the fluid domain. The application of this method has been mainly for the aortic valve [11,109,110].

The main challenge for immersed finite element methods is the coupling between the thin surface ($\Omega_s$) and the fluid domain ($\Omega_f$). Since the fluid and solid domains are discretized with different sets of basis functions, they are inherently incompatible on the interface $\Gamma$. Traditionally, it is required that the continuity of the velocity and stress fields must be strictly enforced. However, this requirement requires perfect-matching moving meshes for both $\Omega_s$ and $\Omega_f$. This requirement poses a severe restriction for the mesh topology [111]. To alleviate such a difficulty, the continuity of the velocity field is suggested to be enforced via a regularization form (augmented Langrangian framework) [112]. In a nutshell, the continuity of the velocity field for the solid $\mathbf{v_s}$ and fluid $\mathbf{v_f}$ across the fluid–solid interface ($\Gamma$) is weakly enforced [110], adding a penalty term to the Lagrangian multiplier on the interface $\Gamma_{fsi}$:

$$\int_{\Gamma_{fsi}} \lambda(\mathbf{v_s} - \mathbf{v_f})d\Gamma + \frac{1}{2}\int_{\Gamma_{fsi}} \beta|\mathbf{v_s} - \mathbf{v_f}|^2 d\Gamma \tag{31}$$

While this method is successfully applied for elastic structures with a finite thickness [112], the zero-thickness condition of the aortic valve leaflet poses a challenge for the penalty term minimization. The choice for $\beta$ plays a significant role in attaining the converged solution. Nevertheless, the immersed framework allows the successful coupling of tissue dynamics and blood flow [113] with the physiological condition [114].

## 6. Particle Methods

Particle methods for heart valves have become more popular over the last decade [78,115–120]. In addition to computational purposes, particle methods could also be used to process the FSI results as a mean of Lagrangian particle tracking for platelet activation [121], flow topology analysis [45], and particle residence time modeling [122]. As the literature in this area is vast, we focus only on one particle method, the lattice Boltzmann method (LBM), in this review.

The LBM is a direct numerical simulations (DNS) method where the continuous fluid phase is modeled as a continuous distribution of fictitious fluid particles. In this regard, the LBM is a mesoscopic method, as it models the distribution of particles rather than the particles themselves. The method is based on kinetic gas theory and the distribution is discretized on a regular three-dimensional lattice grid, where it contains 19 velocity vectors (see Figure 7a for D3Q19 links) with varying lattice weights. The review by Aidun and Clausen [123] provides further details of utilizing the LBM for complex flows. Depending on the accuracy and dimension of the problem, the number and schema of the discretization vector can change. In this method, the distribution function changes in time through a two-step process known as streaming and collision (see Figure 7b), where particles move to neighboring nodes and collide with the existing fluid particles. So, a complete description of the calculations only requires the information in neighboring nodes and, as such, the calculations can be parallelized at an immense scale with minimal scaling loss.

The lattice Boltzmann equation, using the single-relaxation-time Bhatnagar–Gross–Krook (BGK) collision operator, is given as

$$f_i(\mathbf{x} + \mathbf{e_i}, t + 1) = f_i(\mathbf{x}, t) + \frac{1}{\alpha}(f_i(\mathbf{x}, t) - f_i^{(0)}(\mathbf{x}, t)) \tag{32}$$

where $f_i$ is the distribution function, **x** is the fictitious particle location, $\mathbf{e_i}$ is the lattice unit vector, $t$ is time, $\alpha$ is relaxation time, and $f^{(0)}$ is the equilibrium distribution function. Here, the index $i$ represents each velocity vector, which takes 0 to 18 in D3Q19 model. The equilibrium distribution function is given by

$$f_i^{(0)}(\mathbf{x}, t) = \rho w_i \left[ 1 + \frac{1}{c_s^2} (\mathbf{e_i} \cdot \mathbf{v_f}) + \frac{1}{2c_s^4} (\mathbf{e_i} \cdot \mathbf{v_f})^2 - \frac{1}{2c_s^2} (\mathbf{v_f} \cdot \mathbf{v_f}) \right] \tag{33}$$

where $\rho$ is density, $c_s$ is pseudo-sound speed, and $\mathbf{v_f}$ is continuous phase velocity. In the D3Q19 model, the lattice weights, $w_i$, satisfy $\sum w_i = 1$; they are 1/3, 1/18, and 1/36 for the rest, nondiagonal, and diagonal directions, respectively. The macroscopic flow properties (continuous phase), density, velocity, and pressure are obtained using moments of equilibrium distribution, which are given as,

$$\rho = \sum_i f_i^{(0)}(\mathbf{x}, t) \tag{34}$$

$$\rho \mathbf{v_f} = \sum_i f_i^{(0)}(\mathbf{x}, t) \, \mathbf{e_i} \tag{35}$$

$$c_s^2 \rho \mathbf{I} + \rho \mathbf{v_f} \otimes \mathbf{v_f} = \sum_i f_i^{(0)}(\mathbf{x}, t) \, \mathbf{e_i} \otimes \mathbf{e_i} \tag{36}$$

respectively, where **I** is the identity tensor. Here, the pressure is proportional to the density by $p = c_s^2 \rho$.

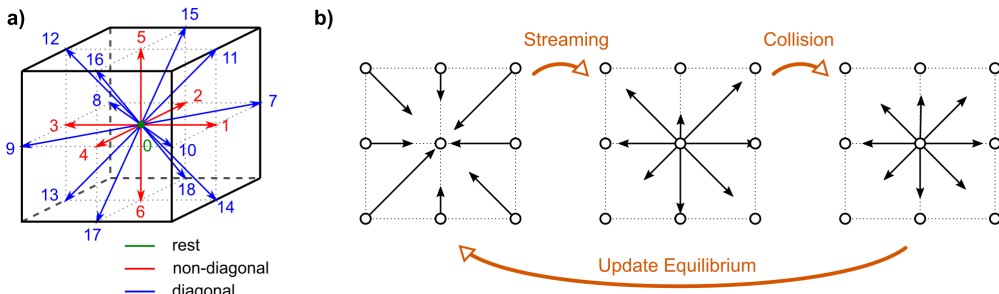

**Figure 7.** Illustration of (**a**) three-dimensional discretization scheme, namely D3Q19 and (**b**) representative process of streaming and collision in two-dimensional discretization [61].

Although the LBM is inherently a direct numerical simulation method, which solves all scales of fluid motion with no turbulence model, at a high Reynolds number it experiences numerical instabilities due to the large spatial gradients of the velocity. To eliminate these instabilities, Keating et al. [124] proposed an entropic lattice Boltzmann (ELB) method that applies a discrete H-theorem constraint to enforce universal positive-definiteness on all particle distribution values. The ELB equation is derived from the single-relaxation LBM equation, and is written as

$$f_i(\mathbf{x} + \mathbf{e_i}, t + 1) = f_i(\mathbf{x}, t) + \frac{\gamma}{2\alpha} (f_i(\mathbf{x}, t) - f_i^{(0)}(\mathbf{x}, t)) \tag{37}$$

where $\gamma$ is the non-trivial root of the discrete form of the standard continuum Boltzmann H function. This is a powerful method (highly scalable) for the DNS of low re-transient turbulent flows. It brings detailed balance to the discrete Boltzmann equation and therefore it is unconditionally stable as long as the grid system resolves the small eddies.

The standard bounce back (SBB) of the LBM is derived to govern the fluid–structure interaction (FSI) in both one-way or two-way manners, and the accuracy has been shown by early studies [123,125,126]. It is shown that if the boundary occurs at the midpoint of the link, the SBB becomes a second-order accurate method [127]; whereas in complex geometries the accuracy degrades to first-order [128,129]. To increase the accuracy, two groups of approaches are broadly adopted: link-based methods and node-based methods. For link-based methods, the SBB method is modified to accurately account for the arbitrary boundaries by interpolating post-bounce-back distribution to the nearest fluid boundary node. Here, the interpolation accuracy can be set to second and higher orders. A detailed examination of interpolated link-based methods can be found in Pan et al. [130]. In node-based methods, the fluid nodes adjacent to solid surface are directly altered by modifying the collision term with a mass-conserving force term. These methods, immersed boundary (IB) [82] and external boundary force (EBF) [131], broadly appear as a two-grid system where the solid interface is defined in a Lagrangian grid in addition to the Eularian LB grid. Thus, they provide much smoother force and torque calculations over the surfaces of deformable/nondeformable particles.

In application to the BMHV, the SBB method derived in one-way and two-way FSI couplings provides great accuracy. In this FSI method, as is the case with calculations of the continuous phase, the interaction between the continuous phase and boundaries is calculated locally so that it can also be efficiently parallelized. In one-way coupling, the prescribed motion of the leaflets imparts force to the fluid phase, whereas, in two-way coupling, the pulsation of the flow exerts forces to leaflet surfaces as well. Yun et al. [132] conducted a study to compare the accuracy of one-way and two-way coupling FSI. In one-way coupling, the angular position of the leaflets is prescribed from an experimental study by Dasi et al. [21]. In two-way coupling (Equation (25)), forces across the leaflet surface are calculated and a Newtonian angular dynamics equation is solved by allowing only the rotational motion about the hinge fulcrum line (Equation (4)), while friction in the hinges is neglected and a moment of inertia for each leaflet is included. Their findings indicate a good agreement between experimental and two-coupling leaflet angular positions. This shows the potential of using a two-way coupling FSI if the prescribed motion information is not available.

In accurately calculating the boundary interactions of small particles, i.e., platelets, the external boundary force (EBF) method is derived [131,133]. Instead of using the SBB method, which assumes that the solid boundary is always located halfway between lattice links, the EBF method creates a Lagrangian frame inside the Eularian fluid phase and it calculates interactions at the exact location of the particle surface. Then, for the fluid phase, EBF is added as an additional force in to the LBM equation, and for particle motion and orientation, Newtonian dynamics equations are solved. One major advantage of this method over SBB is that it can accurately calculate fluid properties, such as wall shear stress on the particle surfaces (see Figure 8b). This ability provides a unique potential over previous studies, where platelets are considered in a point-like manner, with no volume or surface, in determining platelet activation and blood damage.

At this point, it is worth mentioning that, with the Newtonian LB method, such an accurate modeling of particles (platelets, RBCs, etc.) suspended in Newtonian fluid (blood plasma) provides an accurate description of the non-Newtonian characteristics of the suspension. In addition to this, the LB method can be derived to govern the shear-thinning rheology of the blood flow [134]. Starting with the power-low model, the shear-rate-dependent apparent viscosity can be described as

$$\mu = k(\dot{\gamma})^{n-1} = \mu_0(\sqrt{2D})^{n-1} \qquad (38)$$

where $k$ is consistency index, $\dot{\gamma}$ is shear rate, $n$ is the degree of non-Newtonian behavior, and $D$ is the second invariant of the rate-of-strain tensor. When the apparent viscosity is calculated based on this relation, local relaxation times for all lattices are determined. For further details about this approach and stable solutions, which are subject to the implementation of the truncated power-law model and tuning relaxation, readers are referred to the study by Gabbanelli et al. [135].

As described above, improving the numerical stability of the standard LBM with entropic LB and integrating fluid–structure interactions and the external boundary force method provides an extensive approach to describe and solve complex multiscale flow dynamics through the BMHV. Figure 9 shows vorticity field and $Q$-criterion iso-surfaces during critical key points at a given cardiac flow cycle. The high spatiotemporal resolution captures the critical flow characteristics from the opening phase to the leakage phase, such as the formation of a b-datum upstream jet, the interactions between the leaflet downstream von Karman street and vortex structures formed by sinus expansion, and the vortex washout in the downstream. Such an accurate description of flow dynamics through a cardiac cycle is also essential in detecting hemolysis and shear-induced platelet activation leading to thrombogenesis.

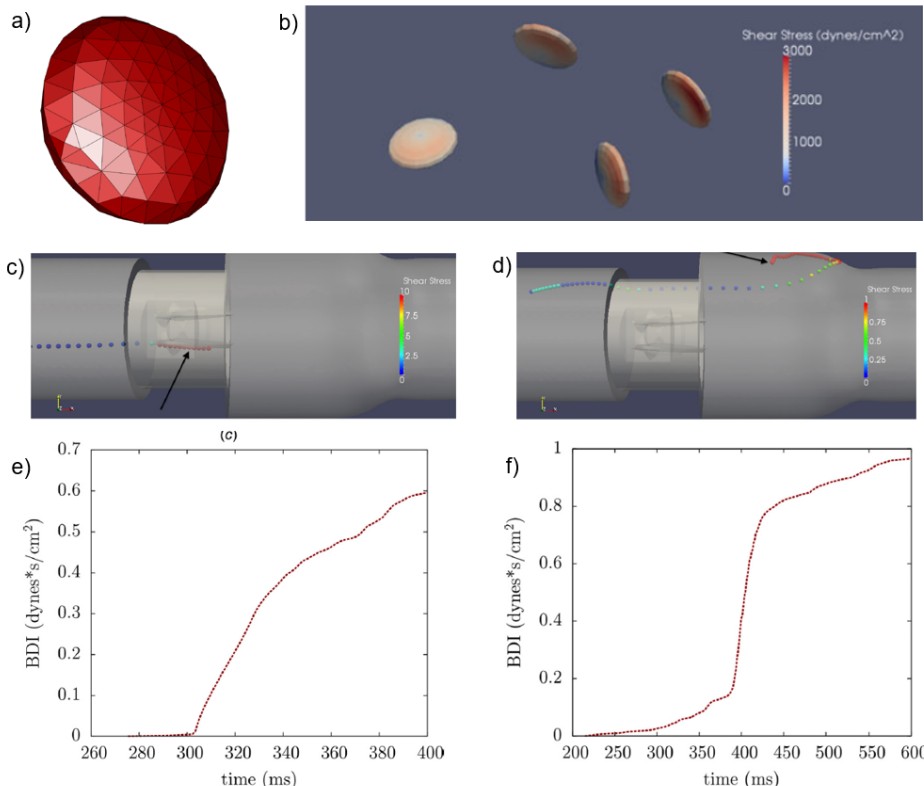

**Figure 8.** Visualization of blood damage index analysis. (**a**) Modeled platelet with surface mesh of 292 triangular elements, 3 μm major axis diameter, 1.3 μm minor axis diameter. (**b**) Platelets in flow through a BMHV colored by instantaneous surface shear stress magnitude. Perpendicular viewpoints of platelet pathline while (**c**) traversing near leaflet and (**d**) platelet caught in recirculation near sinus expansion wall, (**e,f**) corresponding damage accumulation overtime, respectively. Reprinted by permission from the *Journal of Biomechanical Engineering* [136].

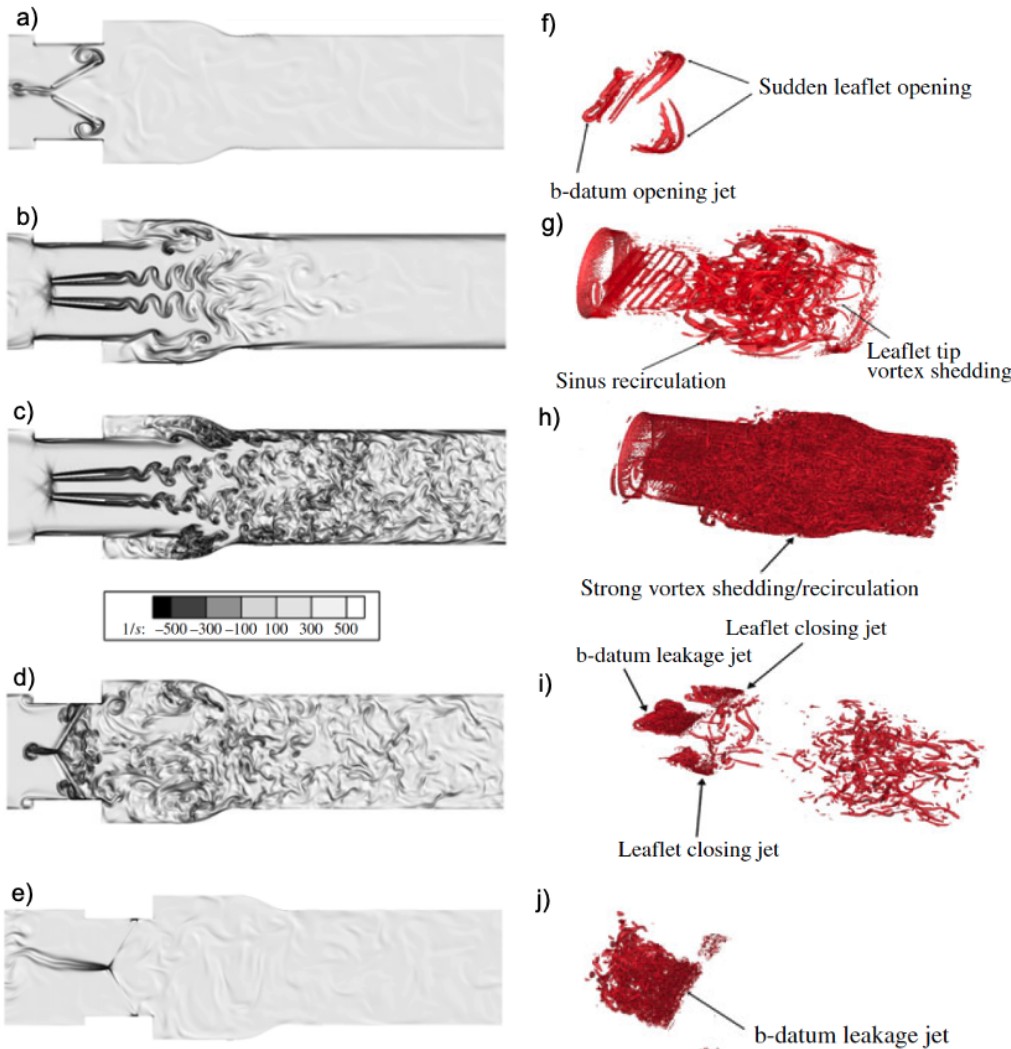

**Figure 9.** Pulsatile flow visualization of three-dimensional BMHV simulations. (**a–e**) Vorticity field and (**f–j**) *Q*-criterion for (**a,f**) opening phase, (**b,g**) acceleration phase, (**c,h**) peak flow, (**d,i**) closing phase, and (**e,j**) leakage phase. Reprinted by permission from the *Journal of Fluid Mechanics*, Cambridge University Press [137].

Blood damage index can be defined based on a simple shear stress exposure time relation defined by Dumont et al. [138],

$$\text{BDI} = \frac{1}{n} \sum_{i=1}^{n} \tau_i \cdot \Delta t_i \tag{39}$$

where BDI is blood damage index, $n$ is the number of platelets, $\tau_i$ is the maximum surface shear stress, and $\Delta t_i$ is the exposure time. Thanks to the LBM-EBF method, which accurately calculates local wall shear stress on the platelet surface (see Figure 8b), BDI is computed based on the maximum shear stress determined on meshed platelet surfaces. Min Yun et al. [136] simulated pulsatile flow through a 23 mm St. Jude Medical (SJM) regent valve. Figure 8c–f shows the trajectory of a platelet and the corresponding BDI over time. With this highly resolved multiscale simulation, they confirmed that although no platelets exceeded the activation threshold, significant damage occurred for suspended platelets trapped in re-circulation zones. Moreover, Yun et al. [132] utilized the LBM-EBF method to investigate the BDI index of three different BMHV hinge designs. Their numerical study predicted a consistently higher BDI index for the CarboMedics (CM) aortic valve hinge,

which agrees with the experimental studies. Hence, it shows the capability of the LBM-EBF method in guiding the design of the BMHV.

## 7. Current Challenges and Future Directions

One main challenge in patient-specific modeling for the left heart is the dynamic nature of the geometry $\Gamma_{LV}$. With the advance of machine learning algorithms, various groups [139] have proposed an automated process for delineating the LV geometrical deformation for hemodynamic simulations [140]. In addition, recent developments in ultrasound techniques [70,141] and 4D CT scans [142] can provide higher temporal and spatial resolutions for $\Gamma_{LV}$ dynamics as well as the intraventricular flows. However, it is expected that the progress in this front might be limited by the physiological constraints of human subjects.

The success of FSI models for heart valves [3] has led to recent efforts in coupling blood flow dynamics with multi-physic processes in the human heart, such as endocardial dynamics, acoustics, chemical transport, and electrodynamics. Biochemical mass transport processes play an important role in heart pathology, such as thrombosis and leaflet calcification, thus advection–diffusion-reaction models in an FSI environment have recently been used to study these processes [45]. Doctors have been using stethoscopes to diagnose heart diseases because abnormal heart sounds are associated with various conditions [143]. This association has motivated the study of the sound generation process (acoustics) [144] in valvular murmurs. Furthermore, attempts have been made to couple electro-mechanical models of heart tissues with blood flows [7,31]. However, current computational models require the inputs from non-invasive modalities, such as the MRI, which has limitations in both spatial and temporal resolutions, as discussed in Section 4. Hence, a complete model for the human heart and its valves is still an important goal to be attained.

Besides the above-mentioned methods, other alternative approaches could contribute to new developments of computational methods for heart valves. For example, smooth particle hydrodynamics have recently been applied for heart valves [78,115–120]. As particle methods are well suited for graphics processing units (GPUs), it is possible that future works might utilize this advantage, as computing platforms have shifted their power toward GPU-accelerated simulations.

New tools in machine learning and data analytics have opened up a new field in cardiovascular flows, and in valvular flows in particular [145]. In addition, the current development in data analytics promises a new direction in integrating multi-modality data [146] with high-fidelity simulations [147]. Furthermore, the invention of physics-informed neural networks (PINNs) [148] now enables many inverse problems to be solved, such as determining the blood pressure field or flow field directly given only the anatomical geometries [149]. The PINNs approach is at the very early stage of being explored in cardiovascular flows and their clinical application in complex patient-specific settings has yet to be demonstrated.

In this work, the essential components for left heart hemodynamics are briefly reviewed. Recent developments in numerical algorithms for computing the kinematics of valvular leaflets in the human left heart are summarized, while major achievements have been attained with the available computational methodologies, and new trends in data analytics and machine learning have emerged. The fusion of high-fidelity numerical simulation and in vivo data is likely to take place given the inherent constraints in imaging technologies. The major shift in computational methodologies for intraventricular flows might require innovations on multiple fronts.

**Author Contributions:** Conceptualization, F.S., C.A., A.Y. and T.B.L.; methodology, T.B.L.; software, T.B.L.; validation, T.B.L. and C.A.; formal analysis, T.B.L.; investigation, T.B.L.; resources, T.B.L.; data curation, T.B.L.; writing—original draft preparation, T.B.L., F.S. and M.U.; writing—review and editing, F.S., T.B.L., M.U. and C.A.; visualization, T.B.L.; supervision, F.S. and A.Y.; project administration, T.B.L.; funding acquisition, T.B.L. All authors have read and agreed to the published version of the manuscript.

**Funding:** The lead author (Trung Bao Le) receives financial support from the ND-ACES project (NSF #1946202). The computational resources are provided by the North Dakota State University CCAST, which is enabled by the NSF MRI # 2019077.

**Institutional Review Board Statement:** Not applicable.

**Informed Consent Statement:** Not applicable.

**Data Availability Statement:** Not applicable.

**Acknowledgments:** The lead author Trung Bao Le would like to thank the ND EPSCoR Office for providing seed funding for this work.

**Conflicts of Interest:** The authors declare no conflict of interest.

## Abbreviations

The following abbreviations are used in this manuscript:

| | |
|---|---|
| ALE | Arbitrary Lagrangian–Eulerian |
| BMHV | Bi-leaflet Mechanical Heart Valve |
| BGK | Bhatnagar–Gross–Krook |
| BDI | Blood Damage Index |
| CT | Computed Tomography |
| DNS | Direct Numerical Simulation |
| ELB | Entropic Lattice Boltzmann |
| EBF | External Boundary Force |
| FSI | Fluid–Structure Interaction |
| IBM | Immersed Boundary Method |
| LV | Left Ventricle |
| LVOT | Left Ventricle Outflow Tract |
| LCA | Left Coronary Artery |
| NCA | Non-Cusp Coronary Artery |
| LBM | Lattice Boltzmann Method |
| MRI | Magnetic Resonance Imaging |
| PSM | Patient-Specific Modeling |
| RCA | Right Coronary Artery |
| SIB | Sharp-Interface Immersed Boundary |
| SBB | Standard Bounce Back |

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
