# Peer review of "Computational Methods for Fluid-Structure Interaction Simulation of Heart Valves in Patient-Specific Left Heart Anatomies"

_fluids, doi:10.3390/fluids7030094_

Round 1

Reviewer 1 Report

This is an interesting and well written review paper by some of the leaders in the field about FSI modeling of heart valves. The paper nicely focuses on numerical algorithms and provides a concise review and introduction to these models. I only have a few minor comments to further strengthen the paper.

Comments:

  • 1- An interesting and challenging aspect of FSI modeling is multiphysics modeling. This is also a focus of the current MDPI special issue (based on the online description) as mentioned: “models that couple flow with structural dynamics, aeroacoustics, chemistry, electrodynamics, heat transfer, and other physical domains.” Interestingly, some of these physics are relevant to heart valves and pose additional challenges for FSI modeling. These aspects seem to fit with the scope of the special issue and paper and therefore I recommend that the authors briefly discuss them:

1a- Chemistry:  Biochemical mass transport processes play an important role in thrombosis as well as the initiation phase of calcification. For example, advection-diffusion-reaction models in an FSI environment have recently been used to study these processes (Ref 47 in the paper; Sadrabadi et al).

1b- Aeroacoustics: This is relevant to murmur generation in diseased valves. See work done by Mittal group (e.g., Zhu et al. Journal of Biomechanical Engineering 2019).

  • 2- In section 7, it is mentioned that PINNs have not been applied to cardiovascular flows. Please see Raissi et al. Science 2020 and Arzani et al Physics of Fluids 2021. The better way to phrase this sentence is to mention that PINNs are at the very early stage of being explored in cardiovascular flows (the two references above) and their clinical/practical utility in complex patient-specific settings is yet to be demonstrated.
  • 3- In Sec. 6, particle methods (LBM) are discussed for FSI modeling of blood flow. It might be noteworthy to also mention that particle methods could also be used in an FSI setting to post-process the data produced by FSI models of heart valves (Lagrangian particle tracking). Some examples in heart valves include: platelet activation modeling (Danny Bluestein’s work among others), flow topology/physics analysis (Ref 47 and Shadden et al, Chaos 2010), and particle residence time modeling (Mutlu et al., Fluids 2021).   

Author Response

The authors would like to thank the reviewers for spending time with our manuscript and providing us the encouraging comments. These comments have helped the authors to improve the manuscript. Please find our responses in the attached file.

Reviewer 2 Report

Review of  “COMPUTATIONAL METHODS FOR FLUID-STRUCTURE INTERACTION SIMULATION OF HEART VALVES IN PATIENT-SPECIFIC HEART ANATOMIES” by Le et al.

The paper is generally well written and covers many of the aspects relating to fluid-structure interactions. However, much of the background material is fairly standard and can be found elsewhere. There are a number of examples of simulations, but there are no clear conclusions, other than illustrating that the approach works and a small glimpse of some of the results from other papers.  Below are a list of suggested correction, mainly taken from the LBM section which is most closely aligned with my area of expertise.

  1. Line 31: “Therefore, only two valves are of interest: the mitral and the aortic valve.” This should be made clear in the title and the abstract.
  2. Line 32: The reader is refereed to [9] for a review of other aspects of the heart. This paper is over 10 years old. Is there a more recent reference that can be given?
  3. Line 332-335. Although the LBM has its root in the discrete particle modelling of the lattice gas model, it is not truly a particle model ad does not really model fictitious fluid particles.
  4. Line 355. The weighting functions are given for diagonal and non-diagonal directions, but it is not clear which link are on a diagonal for the D3Q19. Also w_0 is not given.
  5. Figure 7: There is a formatting issue with the arrows.
  6. Equations (35) and (36). Here the velocity is given in equation (35) as c_d. but as v_f elsewhere. Not clear how this relates to the pressure as suggested in line 335.
  7. Line 369: Here the authors mention the SBB boundary conditions, but they do not explain what they are. Also, there are a number of alternative methods which provide a second-order accuracy in the complex geometries considered here. It would be important to include these.
  8. The authors state that only Newtonian flows will be considered. Given that the LBM is discussed in some detail, and that two of its main advantages for this type of work are simplicity with which complex boundaries can be modelled (see point 7 above) and non-Newtonian fluids can be simulated, it would be worth including some details about non-Newtonian modelling (at least in the LBM section)
  9. References 9 and 10 are to the same paper.

Author Response

(The authors gave the same response as above.)
